# A Disintegrin and Metalloproteinase with Thrombospondin Motifs 4 Regulates Pulmonary Vascular Hyperpermeability through Destruction of Glycocalyx in Acute Respiratory Distress Syndrome

**DOI:** 10.3390/ijms242216230

**Published:** 2023-11-12

**Authors:** Makiko Konda, Masahiro Kitabatake, Noriko Ouji-Sageshima, Rei Tonomura, Ryutaro Furukawa, Shota Sonobe, Chiyoko Terada-Ikeda, Maiko Takeda, Masahiko Kawaguchi, Toshihiro Ito

**Affiliations:** 1Department of Immunology, Nara Medical University, Kashihara 6348521, Japan; 2Department of Anesthesiology, Nara Medical University, Kashihara 6348521, Japan; 3Department of Thoracic and Cardiovascular Surgery, Nara Medical University, Kashihara 6348521, Japan; 4Department of Diagnostic Pathology, Nara Medical University, Kashihara 6348521, Japan

**Keywords:** ARDS, ADAMTS4, endothelial cells, glycocalyx, vascular permeability

## Abstract

Acute respiratory distress syndrome (ARDS) has no specific and effective treatment, and there is an urgent need to understand its pathogenesis. Therefore, based on the hypothesis that molecules whose expression is upregulated in injured pulmonary vascular endothelial cells (VECs) are involved in the pathogenesis of ARDS, we conducted a study to elucidate the molecular mechanisms and identify target factors for treatment. Primary human lung microvascular endothelial cells (HMVEC-Ls) were stimulated with lipopolysaccharide (LPS) or poly (I:C) and analyzed via a microarray to identify target genes for ARDS. We found that a disintegrin and metalloproteinase with thrombospondin motifs 4 (ADAMTS4) was induced in murine lung VECs in an LPS-mediated ARDS model. Elevated ADAMTS4 was also observed by the immunostaining of lung samples from ARDS patients. The suppression of ADAMTS4 by siRNA in VECs ameliorated LPS-stimulated vascular permeability. The impairment of the cell surface expression of syndecan-1, a marker of the glycocalyx that is an extracellular matrix involved in vascular permeability, was dramatically inhibited by ADAMTS4 suppression. In addition, the suppression of ADAMTS4 protected against LPS-induced reductions in syndecan-1 and the adherens junction protein vascular endothelial cadherin. These results suggest that ADAMTS4 regulates VEC permeability in ARDS and may be a predictive marker and therapeutic target for ARDS.

## 1. Introduction

Acute respiratory distress syndrome (ARDS) is a severe lung disease characterized by pulmonary edema and increased vascular permeability, leading to progressive respiratory failure [1]. The current lack of specific and effective treatment options and the worldwide increase in COVID-19 cases leading to ARDS highlight the urgent need for a better understanding of its pathogenesis in order to develop effective treatments. ARDS is usually associated with direct or indirect lung injury followed by extensive pulmonary inflammation and impaired gas exchange, often requiring mechanical ventilation. Mortality rates are as high as 35–50% [2] and improved treatment strategies are urgently needed.

Acute lung injury is caused by dysregulated inflammation [1]. Microbial products or endogenous molecules associated with cell injury bind to pattern recognition receptors on lung epithelial cells, endothelial cells and alveolar macrophages, and activate the innate immune system [3]. For example, Toll-like Receptor 3 (TLR3) initiates a potent anti-viral immune response by binding to double-stranded RNA ligands [4]. Toll-like receptor 4 (TLR4) is activated by lipopolysaccharide (LPS), a component of Gram-negative bacteria, to induce the production of pro-inflammatory mediators, aiming at the eradication of the bacteria [5]. The immune system also produces reactive oxygen species, leukocyte proteases, chemokines and cytokines, which help to neutralize pathogens but may also exacerbate lung injury [6].

In addition to excessive inflammation, another key pathophysiological abnormality in ARDS is the disruption of the pulmonary microvascular barrier due to increased endothelial and epithelial permeability. Endothelial stabilization is responsible for vascular permeability and is mediated by two types of adhesive apparatus: tight junctions (TJs) and adherens junctions (AJs). Glycocalyx, an extracellular matrix (ECM) that covers the apical side of vascular endothelial cells (VECs) is also associated with vascular permeability. During lung injury, the destabilization of these proteins, as well as the destruction of glycocalyx, leads to increased endothelial permeability and alveolar fluid retention [7], which in turn leads to the formation of pulmonary edema.

In the present study, we hypothesized that molecules upregulated in injured pulmonary VECs regulate vascular permeability and ARDS development, with the aim of elucidating the molecular mechanisms of increased vascular permeability and identifying therapeutic targets.

## 2. Results

### 2.1. Stimulation with LPS or Poly (I:C) in Pulmonary VECs Increases ADAMTS4

First, to identify whether molecules upregulated in injured VECs regulate vascular permeability and ARDS pathogenesis, we comprehensively analyzed the gene expressions from the primary human lung microvascular endothelial cells (HMVEC-Ls) stimulated with TLR ligands, LPS or poly (I:C) using DNA microarray technology. DNA microarray analysis showed that inflammatory cytokines, such as interleukin-6 (*IL6*); chemokines, such as *CXCL1*; and markers found in cytokine-stimulated endothelial cells, such as E-selectin (*SELE*) and intercellular adhesion molecule 1 (*ICAM1*), were highly expressed in both LPS- and poly (I:C)-stimulated cells (Figure 1A,B). A total of 930 genes and 1282 genes were increased by more than two-fold in LPS- or poly (I:C)-stimulated cells, respectively (Appendix A). Furthermore, among the genes increased by more than two-fold in LPS- or poly (I:C)-stimulated cells, 122 genes were highly expressed—not only stimulated cells but also unstimulated cells (Expression level > 10). The top 20 genes of the 122 genes are listed in Figure 1A,B. Among top 20 genes, we focused on a disintegrin and metalloproteinase with thrombospondin motifs 4 (ADAMTS4), because ADAMTS4 has been reported to degrade proteoglycans and extracellular matrix proteins in cartilage and brain [8]. Extracellular matrix protein has been implicated in vascular permeability [9], and therefore, we hypothesized that ADAMTS4 might be involved in increased pulmonary vascular permeability, the main pathology of ARDS. Next, we also confirmed via quantitative PCR analysis that the mRNA expression of ADAMTS4 was significantly induced following both LPS and poly (I:C) stimulation, which was consistent with the results of the microarray analysis (Figure 1C). We also investigated the gene expression of ADAMTS4 in two different cell lines of endothelial cells, human umbilical vein endothelial cells (HUVECs) and mouse VEC cell line (UV♀2). The gene expression of ADAMTS4 was increased following LPS and poly (I:C) stimulation in both HUVECs and UV♀2 (Figure 1D,E). In addition, HMVEC-Ls treated with inflammatory cytokines, IFN-β, IL-6 and/or TNF-α, which were elevated following LPS and poly (I:C) stimulation (Figure 1F–H), showed a predominant increase in ADAMTS4; in particular, the gene expression of ADAMTS4 following IL-6 plus TNF-α stimulation was synergistically increased at 1 h (Figure 1I). Although the neutralization of IL-6, TNF-α or IFN-β alone did not significantly suppress the induction of ADAMTS4 by LPS or poly (I:C), the neutralization of IL-6 plus TNF-α partially suppressed the LPS-mediated induction of ADAMTS4 (Figure 1J,K). In addition, the inhibition of NFκB signaling, which was mediated by inflammatory cytokines as well as LPS or poly(I:C) itself, by NFκB inhibitors BAY-11-7082 or SC75741 completely suppressed ADAMTS4 induction by LPS or poly(I:C) (Figure 1L,M).

### 2.2. Elevated ADAMTS4 in Pulmonary VECs in the ARDS-Model Mice and Patients with ARDS

Next, we investigated whether ADAMTS4 is induced in the ARDS mouse model. We previously established an ARDS mouse model by the direct inoculation of LPS intratracheally [10]. In the control group, an equal volume of PBS was inoculated. Histological findings revealed the accumulation of leukocytes in the lung parenchyma and the dense accumulation of leukocytes and pulmonary edema, particularly at 72 h following LPS administration, confirming the successful establishment of the ARDS model. The histological score of ARDS is increased over time until 72 h following LPS administration (Figure 2B). In this ARDS model, *IL6* mRNA expression in whole lungs was increased to the maximum at 6 h after LPS administration and decreased thereafter, but it was still significantly higher compared to the control group even at 72 h following LPS administration (Figure 2C). ADAMTS4 mRNA expression in whole lungs was similarly increased to the maximum at 6 h after LPS administration and decreased thereafter (Figure 2D). In situ hybridization for ADAMTS4 transcripts in ARDS model lung sections showed that ADAMTS4 was expressed along the VECs at 24 h after LPS administration (Figure 2E). We also confirmed that ADAMTS4 was expressed along the VECs by immunohistochemical staining for VEC surface marker CD31 (also known as platelet endothelial cell adhesion molecule 1 (PECAM-1)) using the continuous specimens (Figure 2F). Moreover, to investigate whether ADAMTS4 was enhanced in the lungs of patients with ARDS, immunohistochemistry was performed using lung specimens from ARDS and non-ARDS patients. In ARDS patients, ADAMTS4 deposition was strongly observed in endothelial cells along the pulmonary vascular endothelium, while it was not detected in lungs from non-ARDS patients (Figure 2G). Immunohistochemical staining was performed on three lung specimens from both ARDS and non-ARDS patients with similar results.

### 2.3. ADAMTS4 Is Involved in Increased Vascular Permeability

Next, to examine whether elevated ADAMTS4 in VECs contributes to the regulation of pulmonary vascular permeability, we performed permeability assays using Boyden chambers. HMVEC-Ls, seeded on trans-well inserts, were transfected by control or *ADAMTS4* siRNA and then stimulated with or without LPS. Twenty-four hours later, fluorescein isothiocyanate (FITC)-labeled dextran was added in the upper chamber, and the intensity of leaked fluorescent substance in the lower chamber was measured (Figure 3A). Transfection with *ADAMTS4* siRNA significantly reduced ADAMTS4 mRNA levels, as confirmed by quantitative PCR (Figure 3B). HMVEC-Ls stimulated with LPS significantly increased the fluorescence intensity compared to the unstimulated group, indicating the increased vascular permeability, while pre-treatment with *ADAMTS4* siRNA significantly reduced the fluorescence intensity compared to the control siRNA group following LPS stimulation (Figure 3C).

### 2.4. ADAMTS4 Contributes to the Destruction of Glycocalyx in HMVEC-Ls

To understand the regulatory mechanisms of vascular permeability of ADAMTS4, we investigated the effect of ADAMTS4 on glycocalyx. The surface expression of syndecan-1, which is one of the major and representative components of glycocalyx, was assessed by fluorescence staining using HMVEC-Ls. In the control group, no differences were observed in syndecan-1 staining intensity between control siRNA- and *ADAMTS4* siRNA-transfected cells (Figure 4A). Syndecan-1 staining intensity in cells transfected with control siRNA was markedly reduced by LPS stimulation, whereas *ADAMTS4* siRNA-transfected cells recovered the syndecan-1 staining intensity following LPS stimulation. Moreover, syndecan-1 was decreased following LPS stimulation in control siRNA-transfected VECs at the protein level. In contrast, syndecan-1 was protected in *ADAMTS4* siRNA-transfected VECs (Figure 4B). Importantly, the protein level of an endothelial-specific AJ protein, vascular endothelial (VE)-cadherin, was also preserved in *ADAMTS4* siRNA-transfected VECs but not in control siRNA-transfected VECs following LPS stimulation.

## 3. Discussion

ADAMTS4 has been reported to degrade proteoglycans and extracellular matrix proproteins in cartilage and the brain [8]. As extracellular matrix proteins are involved in vascular permeability [9], ADAMTS4 may be involved in increased pulmonary vascular permeability, which is the main pathology of ARDS, hence the present study. In this study, we found that ADAMTS4 was expressed in the VECs of ARDS patients, as well as the in vitro and in vivo models of ARDS, and the suppression of its expression ameliorated LPS-induced increases in vascular permeability through the regulation of the glycocalyx. ADAMTS4 is a metalloprotease that degrades proteoglycans, such as aggrecan and versican, in the extracellular matrix and is involved in cell–cell and cell–matrix adhesion [8,11,12]. ADAMTS4 is known to be synthesized by arterial smooth muscle cells, endothelial cells and macrophages involved in the formation and progression of atherosclerotic lesions [13] and osteoarthritis [14], and its expression has been reported to be induced by inflammatory cytokines, such as IL-1, TNF-α and IL-6 [15,16,17,18]. Our study showed that the stimulation of the pulmonary VECs by LPS or poly (I:C), a major component of the outer membrane of Gram-negative bacteria and a synthetic analog of double-stranded viral RNA, respectively, induced the expression of inflammatory cytokines and ADAMTS4. Inflammatory cytokines, such as TNF-α, IL-6 and type I IFNs, especially a combination of these cytokines, could induce ADAMTS4 in pulmonary VECs, although the inhibition of ADAMTS4 induction by the neutralization of these cytokines was partial. On the other hand, the inhibition of NFκB signaling, which was mediated by LPS or poly (I:C) itself as well as inflammatory cytokines, was completely suppressed by ADAMTS4 induction, suggesting that ADAMTS4 is induced in VECs by various inflammatory responses synergistically in various types of inflammations, including pathogen infections, contributing to the pathogenesis of ARDS. However, previous reports on the role of ADAMTS4 in ARDS are limited. One study using a mouse model of influenza infection showed that ADAMTS4 gene expression was restricted to fibroblasts and endothelial cells [19]. They clearly showed that ADAMTS4 promotes lethal immunopathology by influenza infection using ADAMTS4-deficient mice. Since the increased expression of ADAMTS4 was observed only in fibroblast at 3 days after influenza infection, they did not assess the contribution of ADAMTS4 on VECs. In our ARDS mice model using LPS, ADAMTS4 was increased within 6 h, and then gradually decreased, suggesting that ADAMTS4 on VECs functions at an earlier phase of inflammation. The main finding of our study is that the increased expression of ADAMTS4 in pulmonary VECs regulate the enhanced vascular permeability in ARDS, and to the best of our knowledge no similar studies have been found.

The glycocalyx is a ubiquitous ECM on endothelial cell membranes, composed of rich proteoglycans linked to glycosaminoglycans such as heparan sulphate and hyaluronic acid [20,21,22]. The glycocalyx is critical for physiological homeostasis, such as vascular permeability, shear stress, the inhibition of leukocyte–endothelial interactions and coagulation [23,24,25,26,27]. Syndecan-1 is expressed on both the basal and apical surfaces of endothelial cells, and the apical surface of syndecan-1 composed the ECM, called the glycocalyx [28]. In endothelial cells, synthesized ADAMTS-4 was localized to endosomes on the cell surface, where it co-localized the ECM component biglycan. In addition, ADAMTS4 is secreted into the extracellular space, and binds to the ECM outside the cells [29]. Inflammatory responses and cell damage in ARDS disrupt and shed glycocalyx components, namely syndecan-1 and heparan sulfate, leading to increased vascular permeability and the development of pulmonary edema [30,31]. Circulating levels of syndecan-1 and heparan sulphate have been considered as clinical biomarkers of glycocalyx damage [32]. We showed that the LPS-mediated production of ADAMTS4 reduced cell surface syndecan-1 and increased the vascular permeability surface, although the reduction in ADAMTS4 in the steady state did not influence syndecan-1 levels. Since it has been reported that ADAMTS4 could directly interact with syndecan-1 on the cell surface [33], ADAMTS4 might cleave syndecan-1 and disrupt the glycocalyx via inflammation, such as in ARDS. In addition, the degradation of versican, which is a known target of ADAMTS4 and associated with syndecan-1 in the glycocalyx, might increase the access to syndecan-1 or the activity of other ECM proteases, such as matrix metalloprotease, leading to a reduction in syndecan-1. Further investigations of how ADAMTS4 alters the composition of the glycocalyx in ARDS are needed.

Our results also showed that the regulation of ADAMTS4 protected the reduction in VE-cadherin in LPS-stimulated VECs. VE-cadherin is required to maintain the integrity of the pulmonary microvascular endothelial barrier [34], and it forms a cis-type dimer and further trans-binds to the cis-type dimer of neighboring cells, thereby contributing to intercellular adhesion and regulating vascular permeability [35,36]. It has been reported that several mechanisms were involved in VE-cadherin degradation in LPS-stimulated endothelial cells [37]. Tyrosine phosphorylation and caveola-mediated endocytosis of VE-cadherin by LPS-TLR4 mediated-signaling was thought to be a major mode of VE-cadherin degradation [38,39]. VEGF also induces VE-cadherin degradation via Src-dependent phosphorylation. In addition, ADAM10 and γ-secretase directly cleaved the extracellular domain of VE-cadherin [40]. Several studies showed that treatment for these targets could control vascular permeability. In our study, ADAMTS4 was immediately induced by LPS and inflammatory cytokines, and regulated VE-cadherin expression. The alteration of glycocalyx composition by ADAMTS4 might change the responsiveness of VEGF and activity of other metalloproteases or directly cleave VE-cadherin. Therefore, the regulation of ADAMTS4 might be a target molecule as an upstream factor of these events, although the detailed regulatory mechanisms of vascular permeability by ADAMTS4 should be clarified.

Although the results of this study provide valuable insights into the pathogenesis of ARDS, there are some limitations. Only female mice were used for ARDS model because male mice have more susceptible variation in body weight, which would result in a variation in the degree of injury. In addition, we did not confirm that the ADAMTS4-mediated degradation of syndecan-1 directly contributed to the increase in vascular permeability. Although the induction of ADAMTS4 on VECs increases vascular permeability, which plays an important role in the pathogenesis of ARDS, it may also play an important role in inflammatory responses and tissue repair. In particular, understanding the contribution of ADAMTS4 to the extravasation of leucocytes, which is essential for eliminating bacteria, may be an important issue related to the pathogenesis of pathogen-mediated ARDS and whether ADAMTS4 induced and contributed to the pathogenesis of ARDS caused by other factors.

Our data show that the inhibition of ADAMTS4 expression can prevent the degradation of endothelial glycans and damage to VE-cadherin. ADAMTS4 is also highly expressed along the vascular endothelium of ARDS patients, providing further evidence that ADAMTS4 is involved in the pathogenesis of ARDS. These findings highlight the importance of glycan integrity in vascular homeostasis and shed light on potential therapeutic strategies aimed at protecting glycans in patients with ARDS. ADAMTS4 may be a potential target for the treatment of ARDS. Further studies should aim to explore the potential of ADAMTS4 inhibitors and develop strategies to protect endothelial glycans. Specifically, it is necessary to explore the involvement of ADAMTS4 in vascular endothelial cells in the pathogenesis of ARDS using in vivo models; ADAMTS4-KO mice are now being generated in our laboratory for a future study. In conclusion, this study provides a new perspective on the molecular pathogenesis of ARDS and paves the way for the development of new biomarkers and therapeutic strategies.

## 4. Materials and Methods

### 4.1. Ethics Statement

All animal experiments performed in this study were approved by The Animal Care and Use Committee at Nara Medical University (Approval No. 12958), and all experiments were performed following the policy of the Care and Use of Laboratory Animals, Nara Medical University and Animal Research: Reporting of In Vivo Experiments (ARRIVE) guidelines. All experiments involving humans in this study were approved by the Ethics Committee of the Nara Medical University (Approval No. 2052).

### 4.2. Human Lung Endothelial Cell Culture

Primary human lung microvascular endothelial cells (HMVEC-Ls) and human umbilical vein endothelial cells (HUVECs) were obtained from Lonza (Basel, Switzerland) and cultured in endothelial growth medium-2 supplemented with human epidermal growth factor (hEGF), human vascular endothelial growth factor (hVEGF), human fibroblast growth factor basic (hEGFb), an analog of human insulin-like growth factor-1 (R3-IGF-1), ascorbic acid, hydrocortisone, fetal bovine serum (FBS), heparin and gentamicin/amphotericin-B (GA) (Lonza). Mouse VECs line UV♀2 cells were obtained from the RIKEN BioResource Research Center (RCB1994, Tukuba, Japan), and cultured in Dulbecco’s modified eagle medium (DMEM; Fujifilm Wako, Osaka, Japan) with 10% FBS (Thermo Fisher Scientific, Waltham, MA, USA), 1 mM glutamine, 100 U/mL penicillin and 100 μg/mL streptomycin (all from Fujifilm Wako). Cells were cultured in a 5% CO_2_ incubator at 37 °C, and cells at passages 3–8 were used for all experiments.

### 4.3. ARDS Animal Model

Female C57BL/6 mice (6–8 weeks old) were purchased from CLEA Japan (Tokyo, Japan). All mice were maintained under specific pathogen-free conditions in the Laboratory Animal Research Center of Nara Medical University. Mice were anesthetized by an intraperitoneal injection of sodium pentobarbital (80 mg/kg) and then inoculated through the upper airway with 2.5 mg/kg LPS (O55:B5; Sigma-Aldrich, St. Louis, MO, USA) diluted in phosphate-buffered saline (PBS; Fujifilm Wako) or 50 μL of PBS for the control group. After 6, 12, 24 or 72 h following LPS administration, mice were euthanized by blood collection from the heart under anesthesia. Samples were stored at −80 °C until analysis.

### 4.4. Microarray Analysis

HMVEC-Ls were stimulated with LPS (10 ng/mL) or poly (I:C) (100 μg/mL; InvivoGen, San Diego, CA). Total RNA was isolated from HMVEC-L cells at 4 h after LPS or poly (I:C) stimulation using an RNeasy mini Kit (Qiagen, Hilden, Germany). For the amplification and labeling of cRNA, total RNA was amplified using the a Low Input Quick Amp Labeling Kit, one-color (Agilent Technologies, Santa Clara, CA, USA) according to the manufacturer’s instructions and labeled with cyanine 3 (Cy3 100 ng of labeled total RNA was reverse transcribed into double-stranded cDNA using poly dT-T7 promoter primer. DNA Microarray was performed by DNA Chip Research Inc. (Tokyo, Japan) using Agilent SurePrint G3 Human GE Microarray 8 × 60 K Ver 3.0 (Agilent Technologies, Santa Clara, CA, USA), which contains 58,201 probes. Intensity values for each scanned feature were quantified using Agilent feature extraction software version 10.7.3.1. Only features judged to be error-free (presence flags) were used, and features that were not positive, not significant, not uniform, not above background, saturated or outliers in the population (marginal and absence flags) were excluded. Normalization was performed using Agilent GeneSpring GX version 11.0.2. (per chip: normalization to 75th percentile shift; per gene: normalization to the median of the whole sample).

### 4.5. Real-Time Quantitative PCR

HMVEC-Ls, HUVECs and UV♀2 were stimulated with LPS (10 ng/mL) or poly (I:C) (100 μg/mL; InvivoGen, San Diego, CA, USA). In addition, HMVEC-Ls were treated with 200 ng/mL of recombinant human IL-6 (PeproTech, Cranbury, NJ, USA), 10 ng/mL of recombinant human TNF-α (PeproTech) or 100 ng/mL of recombinant human IFN-β (PeproTech) for 1, 2 or 4 h. For the neutralization of inflammatory cytokines, HMVEC-Ls were pre-treated with 1 μg/mL of control IgG (MOPC-21) or neutralizing antibodies against IL-6 (MQ2-13A5), TNF-α (Mab11) or IFN-β (IFNb/A1), all of which were purchased from Biolegend Inc. (San Diego, CA, USA), for 1 h, and then stimulated with LPS or poly (I:C) for 4 h. For NF-κB signal inhibition, HMVEC-Ls were pre-treated with 10 μM of NF-κB inhibitors BAY-11-7082 (Sigma-Aldrich) or SC75741 (Selleck, Houston, TX, USA) or equal volume of dimethylsulfoxide (DMSO, Fujifilm-Wako) for 1 h, and then stimulated with LPS or poly (I:C) for 4 h. Total RNA was extracted from VECs using an RNeasy mini Kit (Qiagen). RNA was reverse-transcribed using a High-Capacity cDNA Reverse Transcription Kit (Applied Biosystems, MA, USA) to synthesize single-stranded cDNA. Gene expression was analyzed with TaqMan Fast Advanced Master Mix and TaqMan gene expression assays (Thermo Fisher Scientific) using StepOnePlus (Thermo Fisher Scientific). The expression levels of each target gene were analyzed by the relative ΔΔCT method, and gene expression was normalized to *GAPDH* expression. The following TaqMan gene expression assays for *GAPDH* (Hs03929097 and Mm99999915), *ADAMTS4* (Hs00192708 and Mm00556068), *IL6* (Hs00174131 and Mm00446190), *TNF* (Hs00174128) and *IFNB1* (Hs1077958) were used.

### 4.6. Histological Analysis

The lung lobes removed from ARDS model mice were fixed in 4% paraformaldehyde and embedded in paraffin. Lung tissue was sectioned (1–2 µm thick), deparaffinized with xylene and stained with Mayer’s hematoxylin (Sakura Finetek Japan K.K., Tokyo, Japan) and eosin (Nacalai Tesque Corporation, Kyoto, Japan). The severity of pneumonia was scored (0, absence; 1, mild; 2, moderate; and 3, severe) and classified into (1) interstitial inflammation, (2) alveolitis, (3) pleuritis, (4) bronchiolitis and (5) vasculitis. The final score was calculated as the sum of the individual scores. In addition, 1 point was added for pneumonia, edema or thrombus formation, and 0.5 points for infiltration of more than 10% of the lung area [41].

### 4.7. In Situ Hybridization

The expression of *Adamts4* mRNA in the lungs from the ARDS animal model was visualized using a ViewRNA ISH Tissue Evaluation Kit (Thermo Fisher Scientific) in accordance with the manufacturer’s instructions. Briefly, deparaffinized lung sections were boiled for 10 min and treated with a protease solution for 20 min. After formalin fixation, a ViewRNA TYPE 1 Probe Set (mouse *Adamts4*, VB1-3048069-VC, Thermo Fisher Scientific) against *Adamts4* was used for hybridization with the samples for 2 h. Signals were amplified by hybridization with the PreAmplifier Mix and Amplifier mix, which allows for the specific amplification of the target-specific signal by branched DNA technology, and were then developed using Label Probe 1-AP with FastRed Substrate.

### 4.8. Immunohistochemical Staining

For the visualization of ADAMTS4 detection in human lungs, deparaffinized lung sections were incubated at 105 °C for 20 min in 0.01 M citrate buffer (pH 6.0) for antigen retrieval. To remove endogenous peroxidase, slides were treated with 0.3% hydrogen peroxide (Fujifilm Wako) for 10 min, then incubated with anti-ADAMTS4 antibody (LS-A7773, LSBio, Seattle, WA, USA) at 2 µg/mL in PBS containing 0.05% Tween 20 (Fujifilm Wako) for 45 min. Reactions were visualized using Histofine Simple Stain MAX-PO (R) and a Histofine DAB substrate kit (Nichirei Biosciences, Tokyo, Japan) according to the manufacturer’s instructions. To analyze the distribution of CD31 expression on mouse lungs, deparaffinized lung sections were incubated at 105 °C for 10 min in 0.01 M citrate buffer (pH 6.0) for antigen retrieval. After the blocking of endogenous peroxidase, slides were incubated with anti-CD31 polyclonal antibody (ab28364, abcam, Cambridge, UK) at 1/50 dilution in PBS containing 0.05% Tween 20 and 1% bovine serum albumin (Fujifilm Wako) for 30 min. Reactions were visualized using Histofine Simple Stain MAX-PO (R) and a Histofine DAB substrate kit (Nichirei Biosciences, Tokyo, Japan) according to the manufacturer’s instructions.

### 4.9. Transfection of siRNA

HMVEC-Ls were seeded at a concentration of 1 × 10^5^ cells/well. Cells were incubated for 1 or 2 day to become the confluent monolayer. Next, 10 pmol of non-targeting control siRNA (#D001810-10-50) or *ADAMTS4* siRNA (SMARTpool ON-TARGETplus, #L-003807-00-0005; Horizon Discovery Ltd., Waterbeach, UK) in OptiMEM (Thermo Fisher Scientific) were mixed with Lipofectamine™ RNAiMAX transfection reagent (Thermo Fisher Scientific) and reacted for 5 min at room temperature. Then, the siRNA solution was mixed with cultured medium, and the medium was refreshed. After 24 h incubation, HMVEC-Ls were used for the experiment of LPS stimulation.

### 4.10. Permeability Assay

Boyden chambers were used for permeability evaluation; HMVEC-Ls were seeded on Transwell Inserts (1.0 µm pore; Falcon^®^ Transparent PET Membrane) at a concentration of 1 × 10^5^ cells/well. The cells were cultured for 2 days. After monolayer confluency was confirmed, control or *ADAMTS4* siRNA (Horizon Discovery Ltd., Waterbeach, UK) was transfected. After overnight culturing, cells were stimulated with or without LPS (10 ng/mL) for 24 h. To measure dextran leakage in human HMVEC-Ls, 100 μg/mL of assay solution containing FITC-labeled dextran (Molecular weight 3000, MERCK, Darmstadt, Germany) was added to each upper chamber, and after 20 min of incubation at 37 °C, the fluorescence intensity of the medium from the lower chamber was measured at 485–538 nm using a SpectraMax M2/M2e multi-mode plate reader (Molecular Devices, San Jose, CA, USA).

### 4.11. Immunofluorescent Staining

HMVEC-Ls were seeded on glass coverslips (Lab-Tek chamber slide, Thermo Fisher Scientific) at a concentration of 5 × 10^4^ cells/well. After monolayer confluency was confirmed, control or *ADAMTS4* siRNA (Horizon Discovery Ltd.) was transfected. After 24 h of culture, cells were stimulated with LPS for 24 h. After washing three times with PBS at room temperature, cells were fixed in chilled acetone for 5 min, and then stained by allophycocyanin-conjugated anti-human CD138/syndecan-1 antibody (BioLegend) for 30 min. After washing, cells were mounted with DAPI and Prolong Gold antifade reagent (Thermo Fisher Scientific) and covered with coverslips. A total of six fluorescence images were obtained at random positions for each sample from three independent experiments using an all-in-one fluorescence microscope BZ-X810 (Keyence, Osaka, Japan), and fluorescence intensities were automatically calculated using a hybrid cell-count application in .BZ-X Analyzer software (Keyence).

### 4.12. Simple Western Assays

Cells were lysed in RIPA buffer (50 mM Tris-HCl, 150 mM NaCl, 1% Nonidet P40, 0.5% sodium deoxycholate, 0.1% sodium dodecyl sulfate and protease inhibitor cocktail; Nacalai tesque, Kyoto, Japan) and sonicated by using a Bioruptor II (Sonicbio, Samukawa, Japan). After treatment, the centrifuged supernatant was collected, and the protein concentration was determined using a BCA Assay Kit (Thermo Fisher Scientific). For protein detection, the WES automated Western Blotter (ProteinSimple, San Jose, CA, USA) was used with antibodies against ADAMTS4 (11865-1-AP, Proteintech, San Diego, CA, USA), syndecan-1 (#12922, Cell Signaling Technology, Danvers, MA, USA), VE-Cadherin (#2158, Cell Signaling Technology), and β-actin (A5441, Sigma Aldrich) in the 12–230 kDa WES separation module (ProteinSimple).

### 4.13. Statistical Analysis

Statistical analyses were performed using Prism 8.0 (GraphPad software, La Jolla, CA, USA). The Mann–Whitney U test or one-way analysis of variance was used for comparisons. All data are expressed as mean ± SEM from at least two or three independent experiments.

## 5. Conclusions

In vitro and in vivo models of ARDS, increased ADAMTS4 expression in VECs and the suppression of its expression ameliorated LPS-induced increases in permeability. ADAMTS4 also regulated the glycocalyx core protein syndecan-1 in VECs. These results suggest that ADAMTS4 regulates VEC permeability in ARDS and that ADAMTS4 may be a predictive marker and therapeutic target in ARDS.

## Figures and Tables

**Figure 1 ijms-24-16230-f001:**
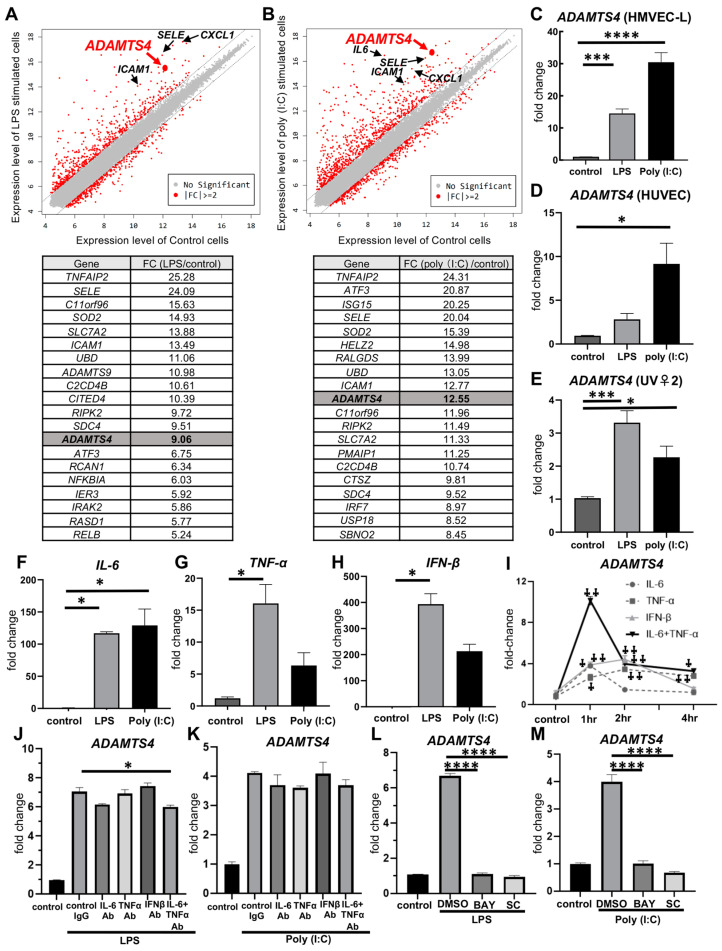
Increased ADAMTS4 expression on HMVEC-Ls following inflammatory stimulation. (**A**,**B**) DNA microarray analysis of HMVEC-Ls stimulated with 10 ng/mL of LPS (**A**) or 100 μg/mL of poly (I:C) (**B**) for 4 h. Scatter plots (upper panel) and the list of the top 20 genes with the highest fold change (FC) (lower panel) compared to unstimulated cells are shown. (**C**–**E**) Quantitative PCR analysis of ADAMTS4 gene expression of HMVEC-Ls (**C**), HUVECs (**D**), and mouse VEC cell line UV♀2 (**E**) following stimulation of LPS or poly (I:C) for 4 h. (**F**–**H**) Quantitative PCR analysis of IL-6 (**F**), TNF-α (**G**) and IFN-β (**H**) gene expression of HMVEC-Ls following stimulation of LPS or poly (I:C) for 4 h. (**I**) Gene expression of ADAMTS4 of HMVEC-Ls stimulated with recombinant IL-6 (200 ng/mL), TNFα (10 ng/mL) and IFN-β (100 ng/mL) for 0, 1, 2 and 4 h. (**J**,**K**) Quantitative PCR analysis of ADAMTS4 gene expression of HMVEC-Ls stimulated with LPS or poly (I:C) for 4 h after pre-treatment with 1 μg/mL of control IgG or neutralizing antibodies against IL-6, TNF-α or IFN-β for 1 h. (**L**,**M**) Quantitative PCR analysis of ADAMTS4 gene expression of HMVEC-Ls stimulated with LPS or poly (I:C) for 4 h after pre-treatment with 10 μM of NF-κB inhibitors, BAY-11-7082 (BAY) or SC75741 (SC), or dimethylsulfoxide (DMSO) for 1 h. Data are shown as the mean ± SEM from two independent explements. * *p* < 0.05, *** *p* < 0.001, **** *p* < 0.0001 ^†^
*p* < 0.001, ^††^
*p* < 0.0001 vs. control.

**Figure 2 ijms-24-16230-f002:**
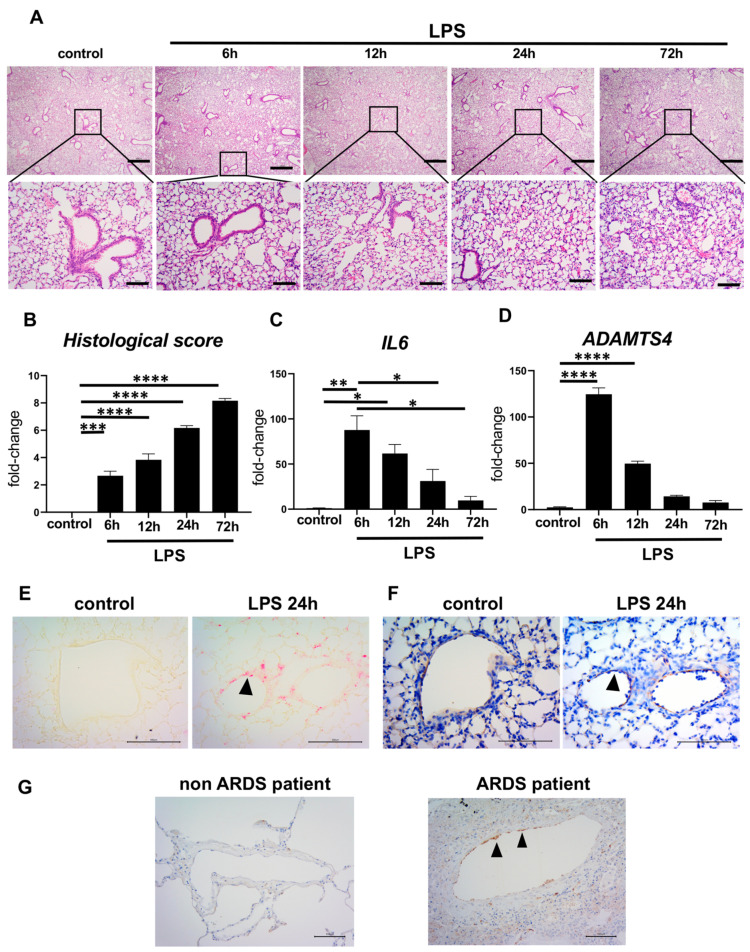
Elevated ADAMTS4 level in ARDS model mice and ARDS patients. (**A**–**D**) C57BL/6 mice (8 weeks old, female) were intratracheally administered with LPS (2.5 mg/kg) and analyzed at 6, 12, 24 and 72 h after LPS administration. (**A**) Representative lung histology by hematoxylin and eosin staining. Magnification = ×40 (top), ×200 (bottom). Scale bar = 500 μm (top), 100 μm (bottom). (**B**) Histological scores, gene expressions of *IL6* (**C**) and ADAMTS4 (**D**) of whole lungs from ARDS model mice. Data are shown as the mean ± SEM (n = 3 in each time point). * *p* < 0.05, ** *p* < 0.01, *** *p* < 0.001, **** *p* < 0.0001. (**E**,**F**) ADAMTS4 mRNA by in situ hybridization (pink) (**E**) and VEC surface marker CD31 by immunohistochemical staining (brown) (**F**) in lungs from control and 24 h after LPS administration. Magnification = ×400. Scale bar = 100 μm. (**G**) Representative immunohistochemistry of ADAMTS4 (brown color) of lung specimens from non-ARDS and ARDS patients (n = 3 each). Arrowheads indicate ADAMTS4-positive cells. Magnification = ×200. Scale bar = 100 μm.

**Figure 3 ijms-24-16230-f003:**
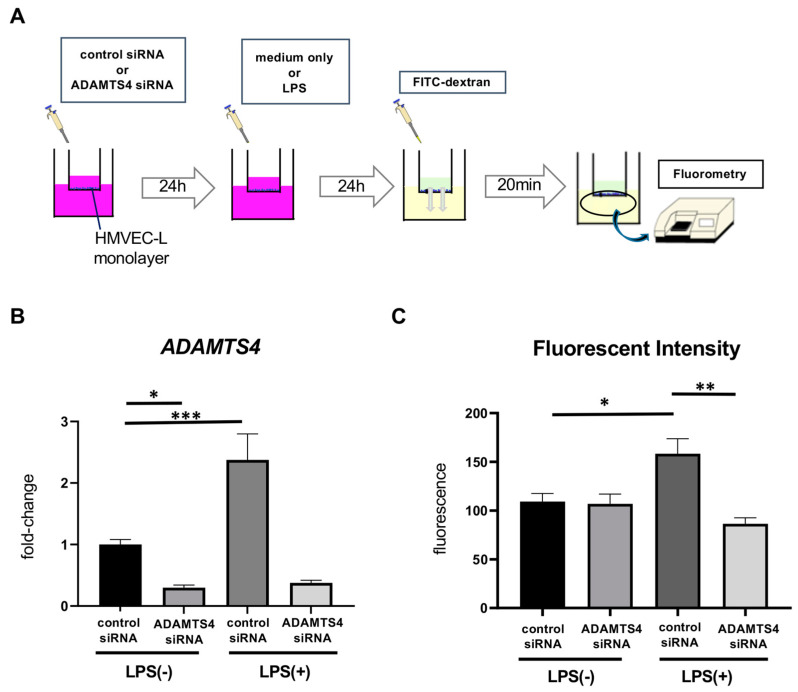
Involvement of ADAMTS4 in increased vascular permeability. (**A**) Schematic procedures for permeability experiments. The details are shown in the Materials and Methods section. (**B**) Expression of ADAMTS4 in HMVEC-Ls stimulated with or without LPS (10 ng/mL) between control and *ADAMTS4* siRNA-transfected HMVEC-Ls. (**C**) Fluorescent intensity of leaked fluorescent substance in lower chamber indicating LPS (1 μg/mL)-induced vascular permeability between control and *ADAMTS4* siRNA-transfected HMVEC-Ls. Data are shown as the mean ± SEM from three independent explements. * *p* < 0.05, ** *p* < 0.01, *** *p* < 0.001.

**Figure 4 ijms-24-16230-f004:**
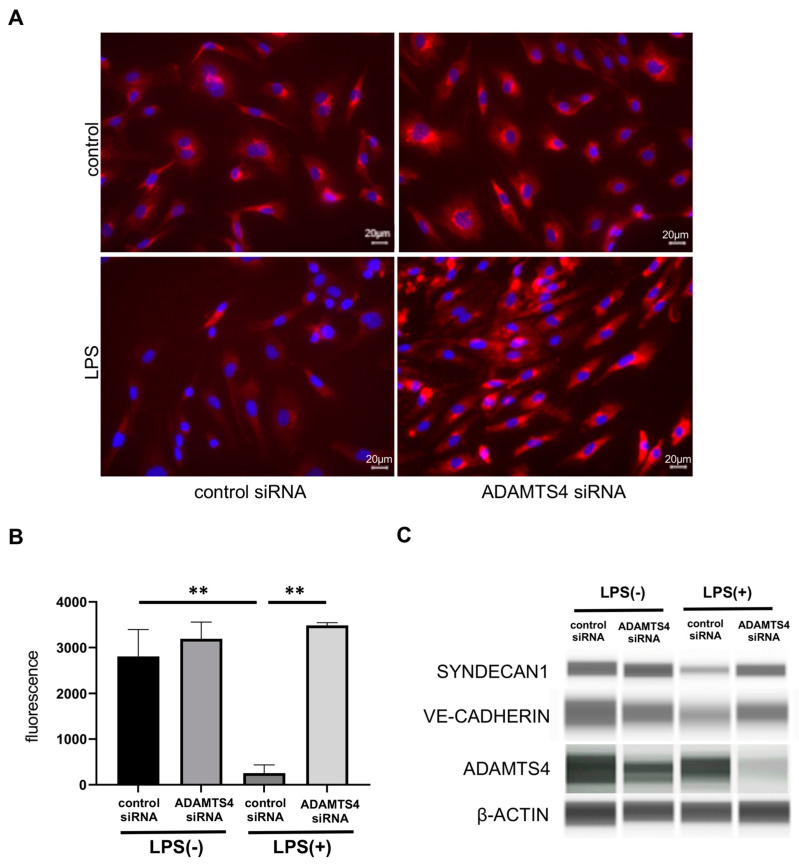
Regulation of glycocalyx damage in vascular endothelial cells by ADAMTS4. (**A**) Fluorescence staining of syndecan-1 following LPS (1 μg/mL) stimulation between control and *ADAMTS4* siRNA-transfected HMVEC-Ls. (**B**) The intensity of syndecan-1 fluorescence following LPS stimulation between control and *ADAMTS4* siRNA-transfected HMVEC-Ls. (**C**) Protein levels of VE-cadherin, syndecan1, ADAMTS4 and β-actin following LPS (1 μg/mL) stimulation between control and *ADAMTS4* siRNA-transfected HMVEC-Ls were determined by the automated protein-blotting system Wes. Data are shown as the mean ± SEM from more than three independent explements. ** *p* < 0.01.

## Data Availability

The datasets used and/or analyzed during this study are available upon reasonable request from the corresponding author.

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
