# Peer review of "A Disintegrin and Metalloproteinase with Thrombospondin Motifs 4 Regulates Pulmonary Vascular Hyperpermeability through Destruction of Glycocalyx in Acute Respiratory Distress Syndrome"

_ijms, 2023, doi:10.3390/ijms242216230_

Round 1

Reviewer 1 Report

Comments and Suggestions for Authors

In this study, Konda et al. investigated that ADAMTS4 was induced in LPS-triggered ARDS model in murine lung vascular endothelial cells (VECs). This induction of ADAMTS4 was also observed in lungs from ARDS. In detail, ADAMTS4 inhibited syndecan-1 to increase permeability of VECs. It is an original work and the findings uncover a novel molecule in VEC permeability in ARDS. However, there are some concerns to be solved.

(1) Since ADAMTS4 plays role in extracellular matrix, the location and distribution with syndecan-1 in VECs need to be clarified.

(2) LPS and poly(I:C) induced IL-6, TNF-α, and IFN-β mediate ADAMTS4 induction. To confirm this, the neutralized assay should be included. In addition, the statistical significance in Figure 1D was absent.

(3) In Figure 3A, the concentration of LPS is displayed as 1 μg/mL, whereas in the figure legend it is 10 ng/mL. What is the final concentration of LPS used?

(4) In Figure 4A, it seems the magnification of cell is not the same. The quality of the images needs to improve.

(5) In Figure 4C, the protein level of ADAMTS4 needs to be verified after siRNA silencing.

(6) Line 73, 74, two-fold and 10-fold can be expressed as the same model.

Comments on the Quality of English Language

The quality of English language is readable.

Reviewer 2 Report

Comments and Suggestions for Authors

In the manuscript entitled “A disintegrin and metalloproteinase with thrombospondin motifs 4 regulates pulmonary vascular hyperpermeability through destruction of glycocalyx in acute respiratory distress syndrome”, the authors investigated the regulation and function of ADAMTS4 in endothelial cells during acute lung injury. Elevated ADAMTS4 was observed in vascular endothelial cells (VECs) in response to LPS or poly (I:C) treatment. The authors also confirmed the upregulation of ADAMTS4 in murine lungs after LPS stimulation and in patients with ARDS. Functionally, the knockdown of ADAMTS4 attenuated LPS-induced permeability via syndecan-1 and cadherin. Although the authors used both in vitro and in vivo experiments, several key questions are not addressed. In addition, the induction of ADAMTS4 in acute lung injury has been reported. Thus, these findings are not novel.

1. Only one in vitro model was used in the entire study and most conclusions are based on HMVEC-Ls cells. To improve the reproducibility and scientific rigors, at least one more type of endothelial cells should be used for validation.

2. Previous study has shown that ADAMTS4 was increased only in lung fibroblasts in influenza-induced lung injury. The authors identified a total of 122 genes upregulated by more than 10-fold, it is unclear why ADAMTS4 was selected for further study.

3. Please add a scale bar in Figure 2A, 2D and 2E.

4. It would be better to show the lung injury score for Figure 2A.

5. How many lung specimens are there from non-ARDS and ARDS patients? No demographic information was provided.

6. Figure 2A, why only female mice were used for experiments?

7. The siRNA knockdown efficiency in HMVEC-Ls needs to be confirmed. There is no description of siRNA sequence and transfection.

8. Although in situ hybridization indicated the increase of ADAMTS4 in murine lungs after LPS administration, it is still unable to demonstrate they are fibroblasts, epithelial cells, or endothelial cells. In this case, double staining with CD31 or other endothelial cell markers may better link in vitro and in vivo data.

9. It would be better to knock down ADAMTS4 in murine lungs after LPS treatment.

10. The material and methods are quite concise. More details should be included.

Round 2

Reviewer 1 Report

Comments and Suggestions for Authors

In this revision, the authors mostly addressed the concerns. However, the data that LPS and poly(I:C) induced IL-6, TNF-α, and IFN-β to mediate ADAMTS4 induction remains unclear. The results need to be improved.

Author Response

Comments 1: In this revision, the authors mostly addressed the concerns. However, the data that LPS and poly(I:C) induced IL-6, TNF-α, and IFN-β to mediate ADAMTS4 induction remains unclear. The results need to be improved.

Response 1: We performed additional experiments of the neutralized assay. Although neutralization of IL-6, TNF-α or IFN-β alone did not significantly suppress the induction of ADAMTS4 by LPS or poly(I:C), neutralization of IL-6 plus TNF-α partially suppressed the LPS-mediated induction of ADAMTS4 (new Figure 1J, K). In addition, inhibition of NFκB singlaing, which was mediated by inflammatory cytokines as well as LPS or poly(I:C) itself, by NFκB inhibitors BAY-11-7082 or SC75741 completely suppressed the ADAMTS4 induction by LPS or poly(I:C) (new Figure 1L, M). These results suggest that ADAMTS4 is induced in VECs by various inflammatory responses synergistically in various types of inflammations including pathogen infections, contributing to the pathogenesis of ARDS. (Line 92-98 & Line 226-229).

Reviewer 2 Report

Comments and Suggestions for Authors

The authors have addressed all the concerns and the quality of the work has been improved. However, only one sex of the mice was used in this study. Thus, please include this point in the Discussion section as a limitation.

Author Response

Comments 1: The authors have addressed all the concerns and the quality of the work has been improved. However, only one sex of the mice was used in this study. Thus, please include this point in the Discussion section as a limitation.

Response 1: Thank you for pointing it out. We included this point in the Discussion section as a limitation. (Line 285-287)

Round 3

Reviewer 1 Report

Comments and Suggestions for Authors

The authors have addressed the issues I concerned. I have no more comments.

Comments on the Quality of English Language

The English is fluent and readable.